# Catecholamines Promote Ovarian Cancer Progression through Secretion of CXC-Chemokines

**DOI:** 10.3390/ijms241814104

**Published:** 2023-09-14

**Authors:** Hyun Jung Kim, Ha Kyun Chang, Yul Min Lee, Kyun Heo

**Affiliations:** 1Department of Biopharmaceutical Chemistry, Kookmin University, Seoul 02707, Republic of Korea; hjkim423@kookmin.ac.kr (H.J.K.); minyzoa@kookmin.ac.kr (Y.M.L.); 2Biopharmaceutical Chemistry Major, School of Applied Chemistry, Kookmin University, Seoul 02707, Republic of Korea; 3Antibody Research Institute, Kookmin University, Seoul 02707, Republic of Korea; 4Department of Obstetrics and Gynecology, Korea University Ansan Hospital, Korea University College of Medicine, Ansan 15355, Republic of Korea; coolblue23@naver.com

**Keywords:** catecholamine, ovarian cancer, invasion, CXCL1, CXCL8

## Abstract

Considerable evidence has accumulated in the last decade supporting the notion that chronic stress is closely related to the growth, metastasis, and angiogenesis of ovarian cancer. In this study, we analyzed the conditioned media in SKOV3 ovarian cancer cell lines treated with catecholamines to identify secreted proteins responding to chronic stress. Here, we observed that epinephrine and norepinephrine enhanced the secretion and mRNA expression of CXC-chemokines (CXCL1, 2, 3, and 8). Neutralizing antibodies to CXCL8 and CXCL8 receptor (CXCR2) inhibitors significantly reduced catecholamine-mediated invasion of SKOV3 cells. Finally, we found that the concentration of CXCL1 and CXCL8 in the plasma of ovarian cancer patients increased with stage progression. Taken together, these findings suggest that stress-related catecholamines may influence ovarian cancer progression through the secretion of CXC-chemokines.

## 1. Introduction

Moderate stress stimulates the sympathetic nervous system, which can have positive effects such as improving concentration, performance, strengthened immunity, and wound healing [1,2]. However, chronic stress that lasts for a prolonged period of time cause various problems including digestive disorders, elevated blood pressure, and decreased immune function, which can contribute to the development of diseases such as heart disease, depression, diabetes, high blood pressure, peptic ulcer disease, obesity, and especially cancer [3,4]. Many studies have shown that chronic stress, which is a situation where psychological stress, continuous worry, and anxiety disorders persist, can promote cancer cell proliferation, metastasis, and angiogenesis, ultimately leading to the progression of various cancers [5,6].

Catecholamines such as adrenaline (epinephrine) and noradrenaline (norepinephrine) are hormones that play a critical role in the body’s “fight or flight” response, which is the physiological response to stress [7]. Catecholamines act on target cells by binding to and activating G protein-coupled receptors (GPCRs), including α-adrenergic receptors (subdivided into α1 and α2 subtypes), beta-adrenergic receptors (subdivided into β1, β2, and β3 subtypes), and dopamine receptors. The activation of these receptors by catecholamines initiates a signaling cascade that involves the activation of G proteins, second messenger systems, and ultimately leads to changes in gene expression and cellular function [8,9,10,11].

While the relationship between chronic stress, catecholamines, and cancer is complex and not fully understood, there is evidence to suggest that chronic stress and the associated release of catecholamines may increase the risk of developing certain types of cancer and promote the growth and metastasis of cancer cells. For example, catecholamines have been shown to stimulate the growth and migration of cancer cells in vitro and in animal models. In colon cancer, epinephrine and isoprenaline, a synthetic catecholamine, have been shown to induce cell proliferation and anti-apoptosis in a xenograft mouse model [12]. Furthermore, norepinephrine enhances ovarian cancer cell aggression and metastasis [13], and epinephrine and norepinephrine promote the growth of ovarian cancers in a xenograft model via the adrenergic receptor pathway [14,15]. Patients taking β-blockers, which block the action of catecholamines such as adrenaline and noradrenaline, have a lower risk of developing certain types of cancer and a better prognosis than patients who do not take β-blockers [16,17]. Additionally, chronic stress has been linked to increased production of chemokines, which can promote the growth and metastasis of cancer cells [6].

Chemokines, which are small peptides (8 to 15 kDa) originally identified as regulators of leukocyte traffic, are classified into four categories (CC, CXC, XC, and CX3C) based on the location of conserved cysteine motifs. Among them, CXC chemokines play pivotal roles in tumor development through both autocrine and paracrine modes. Abnormal expression of chemokines and their specific receptors has been widely implicated in human diseases, particularly autoimmune diseases and cancer. Chemokines correlates with cell migration and cell-to-cell interactions in the tumor microenvironment and influence cancer cell proliferation, invasion, and tumor progression [18,19,20]. Various chemokines exert specific physiological effects in cancer cells by binding to receptors and stimulating downstream signaling pathways. For example, the interaction of receptors in cancer cells with several chemokines, such as CCL2 (MCP-1) and IL-1, induces stimulation of MAPK signaling, leading to the expression of growth-stimulating genes [21,22]. CXCL1 (GRO-α) and CXCL8 (IL-8) promote cancer cell survival by inhibiting apoptosis and consequently inducing upregulation of Mdm2 and downregulation of Bcl-2 expression [23]. In particular, ovarian cancer has been reported to promote angiogenesis and tumor progression through several chemokine-mediated signaling pathways. CXCL1 and CXCL8 chemokines secreted by ovarian cancer cells bind to CXCR1 and CXCR2 receptors and promote cell proliferation, migration, and angiogenesis. Levels of CXCL8 are elevated in the serum of ovarian cancer patients and are associated with poor prognosis and survival. In addition, the overexpression of CXCL8 in ovarian cancer cells activates mitogen-activated protein kinase (MAPK) signaling and signal transducer activator of transcription 3 (STAT3), which in turn increases the expression of VEGF and decreases the level of thrombospondin-1 (TSP-1). Thus, pathways activated by chemokine–receptor interactions appear to be highly involved in the angiogenesis and progression of ovarian cancer cells [23,24,25,26].

Based on previous research, our study aims to investigate the chemokines secreted by catecholamines in ovarian cancer cells and their impact on the progression of ovarian cancer. This area of study is less well defined for ovarian cancer compared to other cancer types. We hypothesize that specific chemokines induced by catecholamines play a role in promoting cancer cell migration and that these chemokines may be elevated in ovarian cancer patients. To test these hypotheses, our study employs various molecular biology techniques to identify catecholamine-induced chemokines and assess their presence in the serum of ovarian cancer patients. Therefore, the results of our study suggest that the relationship between catecholamines and chemokines is closely related to the progression of ovarian cancer.

## 2. Results

### 2.1. Catecholamines Induce Invasion of Ovarian Cancer Cells

To determine which adrenergic receptor subtypes are expressed in human ovarian cancer cells, we examined the mRNA levels of the receptors. All isotypes of adrenoceptors were found to be expressed in HIO-80 (human non-tumorigenic ovarian surface epithelial) and SKOV3 (human ovarian carcinoma) cell lines to varying degrees. However, the α1D and β2 subtypes of the adrenal receptor family were not expressed in the OVCAR3 (human ovarian carcinoma) cell line (Figure 1A). We investigated whether catecholamines could enhance invasion of SKOV3 cells. Epinephrine, norepinephrine, and isoproterenol (an isopropylamine analog of epinephrine) increased the invasion of SKOV3 cells in a dose-dependent manner (Figure 1B).

### 2.2. Catecholamines Induce the Secretion of CXC-Chemokines in SKOV3 Cells

To determine the secretory molecules induced in SKOV3 cells, we harvested the conditioned medium of catecholamine-treated cells and subjected it to SDS-PAG followed by silver staining. We observed several significant protein bands in norepinephrine- or epinephrine-treated samples compared to untreated samples (Figure 2A). Additionally, we performed HPLC analysis of mock- and epinephrine-stimulated conditioned media, which showed several peaks increased in epinephrine-treated conditioned media compared to the control conditioned media (Figure 2B, indicated by asterisks). These data demonstrate that catecholamines activate the adrenergic receptors in ovarian cancer cells, increasing invasion and inducing the secretion of several proteins.

To examine cytokine secretion from SKOV3 cells, we performed an antibody-based protein array assay against 79 human cytokines using either control or epinephrine-containing conditioned media. The results showed that epinephrine upregulates CXCLs (1, 2, and 3 mixed), CXCL1, CXCL8, IGFBP-1, and IL-6 and downregulates TIMP-1 and TIMP-2 (Figure 3).

To evaluate the secretion of each of the CXCL1, CXCL2, CXCL3, or CXCL8, the concentration of each chemokine was examined in catecholamine-treated conditioned media using ELISA or immunoblot analysis. Epinephrine, norepinephrine, and isoproterenol enhanced the secretion of CXCL1 and CXCL8 in both SKOV3 and OVCAR3 cells in a dose-dependent manner (Figure 4A–D). We also observed that epinephrine induced the secretion of CXCL1 and CXCL8 in a time-dependent manner in SKOV3 cells (Figure 4E,F). Since ELISA kits for the detection of CXCL2 and CXCL3 were not available, we confirmed the secretion of CXCL2 and CXCL3 by immunoblot analysis using antibodies specific for CXCL2 and CXCL3. The results showed that the secretion of CXCL2 and CXCL3 was also enhanced with epinephrine treatment (Figure 4G). To determine whether catecholamines activate transcription of CXC-chemokine in ovarian cancer cells, we examined the mRNA levels of each CXC-chemokine by RT-PCR. The mRNA levels of these chemokines were increased by 10 μM of epinephrine at 6 h and decreased at 48 h (Figure 4H). These results demonstrate that catecholamines enhance transcription and consequently increase the secretion of CXCL1, CXCL2, CXCL3, and CXCL8.

### 2.3. CXCL8 Secreted by Ovarian Cancer Cells Induces Cell Invasion

Next, we investigated whether CXCL8 secreted by catecholamines in ovarian cancer cells could induce cell invasion. RT-PCR analysis revealed that all three types of receptors for CXCL1, CXCL2, CXCL3, and CXCL8 (CXCR1, CXCR2, and ACKR1) were expressed in HIO-80, SKOV3, and OVCAR3 cells (Figure 5A). We examined the invasion analysis in the presence of CXCL8 neutralizing antibody or selective CXCR2 antagonists (SB225002 and SB265610). As a result, epinephrine-induced SKOV3 cell invasion was significantly inhibited by CXCL8 neutralization or CXCR2 inhibition (Figure 5B). These results suggest that catecholamine-mediated invasion of ovarian cancer cells is associated with CXCLs secreted by cancer cells.

### 2.4. Elevated Serum Levels of CXCL1 and CXCL8 Are Associated with Advanced Clinical Stages of Ovarian Cancer

To analyze the correlation between CXCL1 or CXCL8 and the clinical stage of ovarian cancer, the concentrations of CXCL1 and CXCL8 were measured in the plasma isolated from ovarian cancer patients using an ELISA. In this study, subjects were divided into samples from healthy subjects (Group 1), patients diagnosed with stage I and II ovarian cancer (Group 2), and patients diagnosed with stage III and IV ovarian cancer (Group 3). The results showed that the levels of CXCL1 and CXCL8 in the plasma of ovarian cancer patients were significantly increased compared to healthy subjects (Figure 6). In addition, it was shown that the levels of CXCL1 and CXCL8 increased as the stage advanced. These results suggest that CXCL1 and CXCL8 are important factors affecting progression and metastasis of ovarian cancer.

## 3. Discussion

Catecholamines secreted by chronic stress are known to induce the onset of various diseases through adrenergic receptor signaling. In some cancers, catecholamines such as epinephrine, norepinephrine, and isoprenaline play an important role. For example, catecholamine activation affects proliferation in colon cancer cells, and β-adrenergic receptor inhibitors effectively suppress proliferation induced by epinephrine and isoprenaline [27]. Adrenergic receptors are upregulated in melanoma tissue, which is the most aggressive type of skin cancer, and norepinephrine and epinephrine promote a number of biological responses [28]. In ovarian cancer xenograft mouse models, chronic stress enhances the tumor growth [13], and epinephrine and norepinephrine lead to increases in metastatic peritoneal tissues in preclinical models of ovarian cancer [29], which are associated with tumor angiogenesis [30]. For these reasons, adrenergic receptors are emerging as attractive targets for cancer therapeutics. Catecholamine secretion within the central nervous system is mediated by cross-talk between chemokines and their receptors. Chemokines induced by the interaction of catecholamines and their receptors are secreted in a paracrine manner to induce various biological effects [31,32].

In the present study, we evaluated the expression of adrenergic receptors in ovarian cancer cells and determined that catecholamines such as epinephrine, norepinephrine, and isoproterenol promote the invasiveness of ovarian cancer cells in a dose-dependent manner. While there are various factors that affect cancer progression, the role of chemokines and chemokine receptors has been well-established by numerous studies. The chemokine system is encoded by nearly 50 human genes, and more than 20 corresponding chemokine receptor genes that are seven-transmembrane G-protein-coupled receptors (GPCRs) [33]. The binding of chemokines to GPCRs facilitates internalization of the receptor, thereby activating various downstream signaling pathways that can lead to the development of various diseases [32]. Among them, chemokines derived from the tumor microenvironment can induce an increase in vascular permeability and tumor cell extravasation [34,35].

Endothelial cells interact by overexpressing their receptors, which activates a feedback loop to induce antigenic support for tumors. For example, endothelial precursor cells (EPCs) express CXCR4, which responds to CXCL12 to migrate to the target tissue, differentiates into EPCs, and forms new blood vessels in mice [36]. This process proceeds when low oxygen stabilizes the transcription factor hypoxia-inducible factor alpha (HIF-1α), causing it to recruit EPCs by upregulating CXCL12 expression [37]. Increased survival and proliferation of cancer cells are associated with chemokine–receptor interactions. Additionally, chemokines secreted by epithelial cells, stromal cells, and normal cells engage receptors that are overexpressed on tumor cells, enhancing the metastatic dissemination of cancer cells. For instance, CXCL12, CCL2, CCL5, and CXCL1 have been shown to regulate growth and stimulate the proliferation of melanoma, glioma, prostate cancer cells, and breast cancer cells [38,39,40,41,42]. Moreover, overexpressed chemokine receptors lead to angiogenesis and promote tumor growth and survival signals for cancer cells.

There is considerable evidence suggesting that chemokines play a significant role in the progression of ovarian cancer. Several studies have identified specific chemokines that are involved in tumor growth, angiogenesis, and metastasis in ovarian cancer. For instance, one study has shown that the chemokine CXCL12, also known as stromal-derived factor-1 (SDF-1), is highly expressed in ovarian cancer and promotes tumor cell proliferation and invasion. Another study found that the chemokine CXCL8, also known as interleukin-8 (IL-8), is overexpressed in ovarian cancer and is associated with poor prognosis and resistance to chemotherapy. In addition to CXCL12 and CXCL8, other chemokines have been implicated in ovarian cancer progression. For instance, the chemokine CCL2, also known as monocyte chemoattractant protein-1 (MCP-1), has been shown to promote tumor growth and angiogenesis in ovarian cancer. The chemokine CXCL1, also known as growth-related oncogene (GRO)-α, has been found to be upregulated in ovarian cancer and promotes tumor growth and invasion.

In the study, we evaluated the expression of adrenergic receptors in ovarian cancer cells and determined that catecholamines, such as epinephrine, norepinephrine, and isoproterenol, promote the invasiveness of ovarian cancer cells in a dose-dependent manner. We hypothesized that catecholamines induce the secretion of chemokines and promote the motility of ovarian cancer cells. We identified CXCL1 and CXCL8 as chemokines induced by epinephrine and confirmed that their secretion in ovarian cancer cells increased in a dose-dependent manner following treatment with epinephrine, norepinephrine, and isoproterenol. Our findings reveal variations in the secretion patterns and quantities of chemokines induced by catecholamines, which appear to be influenced by the histologic type. Ovarian cancer is known for its histological heterogeneity, with various cell lines representing different types. Thus, further studies were needed using ovarian cancer cell lines of diverse histologic types to validate our results. Furthermore, we measured the expression of receptors for CXCL1 and CXCL8 in ovarian cancer cells and demonstrated that specific antibodies, receptor inhibitors, and CXCL8 antibodies SB225002 and SB265610, significantly inhibited ovarian cancer cell invasion induced by epinephrine. We also found that catecholamines induce the secretion of CXCL1 and CXCL8 and increase ovarian cancer cell invasion by interacting with their corresponding receptors, which can be disrupted by specific inhibitors. Moreover, we analyzed the plasma levels of CXCL1 and CXCL8 in ovarian cancer patients at various stages of the disease and observed a significant increase in their levels with disease progression. Therefore, we conclude that the development of targeted molecules to inhibit CXCL1 and CXCL8 may be a promising strategy for suppressing ovarian cancer progression and achieving therapeutic benefits. 

This study has elucidated the correlation between catecholamines, chemokines, and ovarian cancer progression using ovarian cancer cell lines. To further validate the impact of catecholamines and chemokines on ovarian cancer progression, future studies should consider employing xenograft models. Additionally, given the high heterogeneity of ovarian cancer, it is essential to confirm these results using ovarian cancer cell lines representing different histological types.

## 4. Materials and Methods

### 4.1. Cell Lines, Antibody, and Chemicals

The HIO-80 cell line, an immortalized human ovarian epithelial cell line, was obtained from the American Type Culture Collection (ATCC; Rockville, MD, USA) and maintained in a 1:1 mixture of medium 199 and MCDB-105 (Sigma-Aldrich; St. Louis, MO, USA) supplemented with 10% heat-inactivated fetal bovine serum (FBS) (Thermo Fisher Scientific; Waltham, MA, USA) and 0.2 units/mL of insulin (Sigma-Aldrich). The SKOV3 (clear cell) and OVCAR3 (high-grade serous ovarian cancer, HGSOC) cell lines were also obtained from ATCC and maintained in Roswell Park Memorial Institute (RPMI) medium supplemented with 10% FBS.

The Human CXCL8 neutralizing antibody, MAB208, was purchased from R&D systems (Minneapolis, MN, USA). Selective CXCR2 antagonists, SB225002 and SB265610, were obtained from Sigma-Aldrich.

### 4.2. RT-PCR

Total RNA was extracted from ovarian cancer cells using TRIzol reagent (Thermo Fisher Scientific) following the manufacturer’s instructions. To eliminate genomic DNA contamination, total RNA was treated with RNase-free DNase I to eliminate genomic DNA contamination. Complementary DNA (cDNA) was synthesized from total RNA using the GeneAmp kit (Thermo Fisher Scientific). For RT-PCR, 1 μg was amplified for 25 cycles using the primers detailed below. The cycling parameters were as follows: 0.5 min at 94 °C (denaturation), 0.5 min at 55 °C (annealing), and 1 min at 72 °C (polymerization). The products were visualized after electrophoresis on 2% agarose gels containing ethidium bromide. The housekeeping gene GAPDH was used to normalize the gene expression. The primer sequences used in RT-PCR are listed in Table 1.

### 4.3. Cell Invasion Assay

Invasion assays were performed using a 6.5 mm Transwell chamber with an 8.0 μm pore filter (Costar; Cambridge, MA, USA). Each Transwell plate was coated with 1 mg/mL Matrigel (BD Bio Sciences; San Jose, CA, USA), and 5 × 10^4^ SKOV3 cells were seeded into the upper chamber and maintained at 37 °C for 48 h. After incubation, the cells remaining on the upper surface of the chamber were removed using a cotton swab, and the cells that had moved to the lower chamber were fixed and stained with a Diff-Quick staining kit (Sysmex; Kobe, Japan) according to the manufacturer’s protocol. Briefly, invading cells migrated and attached to the bottom of the membrane, and invaded cells were fixed with methanol, cytosol was stained with eosin Y, and the nucleus was stained with methylene blue. An inverted microscope at 100× magnification was used to image the stained cells in five random fields, and the number of cells in each field was manually counted.

### 4.4. High-Performance Liquid Chromatography (HPLC)

HPLC was performed using an Ultimate HPLC system (Agilent 1100; Santa Clara, CA, USA). To achieve high-resolution separation, peptides were separated using a nanoscale RPC analytical column (PepMap C18, 3 mm, 100 Å, 75 μm, id 6150 mm; Agilent Technologies). Mobile phase A consisted of HPLC-grade water containing 0.1% formic acid and mobile phase B consisted of 84% HPLC-grade ACN containing 0.1% formic acid. The separation was performed at a flow rate of 300 nL/min, and the applied gradient was 0–40% of B over 60 min.

### 4.5. Cytokine Antibody Array

To evaluate which cytokines were secreted by catecholamines, 1 × 10^6^ SKOV3 cells were plated in six-well plates and cultured for 24 h. After washing with serum-free RPMI medium three times, the cells were incubated in the absence or presence of 10 μM epinephrine in RPMI medium for 24 h. The supernatants were collected and assayed with a cytokine array kit (RayBiotech; Norcross, GA, USA) containing 79 human cytokine specific antibodies coated on the membrane according to the manufacturer’s instructions.

### 4.6. Silver Staining and Western Blotting

In total, 1 × 10^6^ SKOV3 cells were plated in six-well plates and cultured for 24 h. The medium was then changed to serum-free RPMI medium, and the cells were treated with 10 μM epinephrine or norepinephrine for 24 h. The conditioned medium was obtained, mixed with 5× sample buffer (containing 10% 2-mercaptoethanol), and boiled for 10 min at 95 °C. Protein samples (20 μL) were separated in SDS-PAGE and then the gel was stained by a Silver Staining Kit (Thermo Fischer Scientific) following the manufacturer’s instructions.

For Western blot, 1 × 10^6^ SKOV3 cells were seeded into six-well plates and cultured for 24 h. The cells were treated with 10 μM epinephrine for 0, 3, 6, 12, 24, or 48 h and the conditioned medium (20 μL) was analyzed by Western blot analysis using anti-CXCL2 and anti-CXCL3 antibodies (Cell Signaling Technology, Inc.; Danvers, MA, USA).

### 4.7. Enzyme-Linked Immunosorbent Assay (ELISA)

The levels of CXCL1 and CXCL8 in the culture supernatants of ovarian cancer cells or blood of patients were measured using commercial Quantikine ELISA kits (R&D Systems) according to the manufacturer’s instructions, and 1 × 10^6^ SKOV3 and OVCAR3 cells were seeded into six-well plates and cultured for 24 h. The cells were washed with serum-free RPMI three times and treated with epinephrine, norepinephrine, or isoproterenol. The cell culture media were collected, centrifuged at 50× *g* for 10 min, and analyzed. To isolate plasma, blood samples from ovarian cancer patients were centrifuged at 7000 rpm for 20 min at 4 °C. Absorbance at a wavelength of 450/620 nm was measured with a Synergy H1 microplate reader (BioTek; Winooski, VT, USA). The concentrations of the test samples were calculated based on the standard curve.

### 4.8. Clinical Samples

Following approval by the institutional review boards (IRB number, NCC2018-0104), clinical data from ovarian cancer patients at the National Cancer Center in Korea were analyzed for the present study. Blood samples and individual data were collected from a total of 40 patients diagnosed with epithelial ovarian cancer, fallopian tubal cancer, or primary peritoneal cancer who were undergoing treatment at the same cancer center. Blood samples collected on the day of surgery were immediately refrigerated. Histologic grading was carried out according to the International Union Against Cancer criteria, and the stage of disease was classified according to FIGO staging. Additionally, blood samples were obtained from 10 healthy women aged 20 to 50 as a control group. For the purpose of this study, all histological specimens were reviewed by an experienced pathologist who was blinded to the clinic.

### 4.9. Statistical Analysis

Data were analyzed with GraphPad Prism 8.0 software using one-way analysis of variance (ANOVA) with Dunnett’s multiple comparison test. All data represent the mean ± standard deviation (S. D.). A *p*-value less than 0.05 was considered statistically significant (* *p* < 0.05, ** *p* < 0.01, *** *p* < 0.001, **** *p* < 0.0001).

## 5. Conclusions

The purpose of this study was to investigate the correlation between catecholamines induced by stress and the progression of ovarian cancer at the molecular level. We found that exposing ovarian cancer cells to catecholamines induced the expression of CXCL1 and CXCL8 chemokines, which were shown to promote cancer cell progression in a dose-dependent manner. Our results demonstrated that inhibitors of CXCL8 significantly decreased the progression of ovarian cancer cells, indicating that the development of chemokine inhibitors may effectively suppress the progression of ovarian cancer. Taken together, it has been verified that chemokine secretion increases due to catecholamines induced by stress, and this plays a significant role in the progression of ovarian cancer.

## Figures and Tables

**Figure 1 ijms-24-14104-f001:**
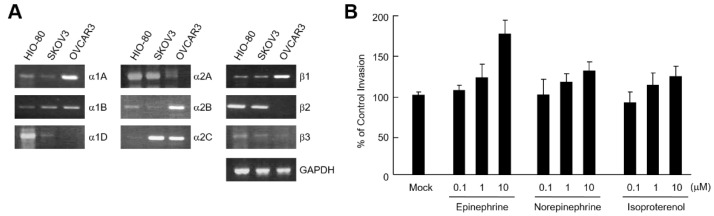
Catecholamines induce invasion of ovarian cancer cells. (**A**) mRNA level of adrenergic receptors in ovarian cancer cells. Total RNA isolated from HIO-80, SKOV3, and OVCAR3 cells was analyzed by RT-PCR using human adrenergic receptors and GAPDH-specific primers. (**B**) 5 × 10^4^ SKOV3 cells were seeded in the upper chambers, with the lower chambers containing 0.1, 1, or 10 μM epinephrine, norepinephrine, or isoproterenol. After culturing for 24 h, the infiltrated cells were fixed, stained, and observed using an optical microscopy. Results are representative of two independent experiments.

**Figure 2 ijms-24-14104-f002:**
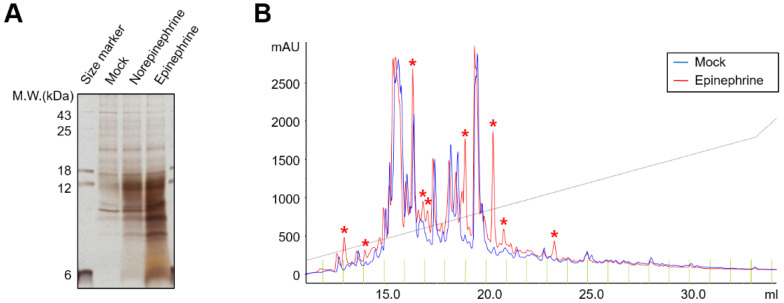
Catecholamines activate protein secretion in SKOV3 cells. (**A**) PBS-, 10 μM norepinephrine-, or 10 μM epinephrine-treated SKOV3 culture supernatants were collected and proteins were separated by SDS-PAGE and visualized by silver staining. (**B**) PBS- or 10 μM epinephrine-treated SKOV3 culture supernatants were collected and performed using an Ultimate HPLC system. Total ion chromatograms of control (Mock, blue) and 10 μM epinephrine (Epi, red) treated culture medium. Peaks specifically increased by epinephrine-treatment are marked with asterisks.

**Figure 3 ijms-24-14104-f003:**
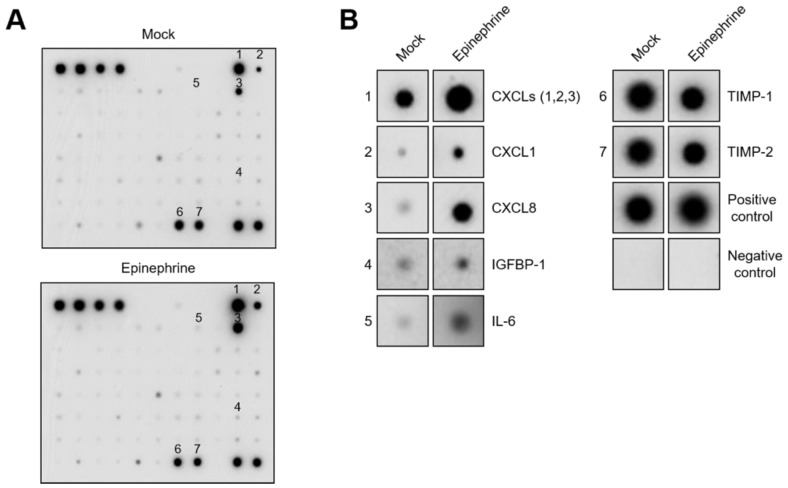
Epinephrine induces CXC-chemokines in SKOV3 cells. (**A**) Levels of cytokines and chemokines in the SKOV3 culture media were determined with cytokine arrays after the cells were treated with vehicle control (top panel) and 10 μM of epinephrine (bottom panel) at 24 h. (**B**) Enlarged images of negative control, positive control, and significantly increased cytokines from (**A**).

**Figure 4 ijms-24-14104-f004:**
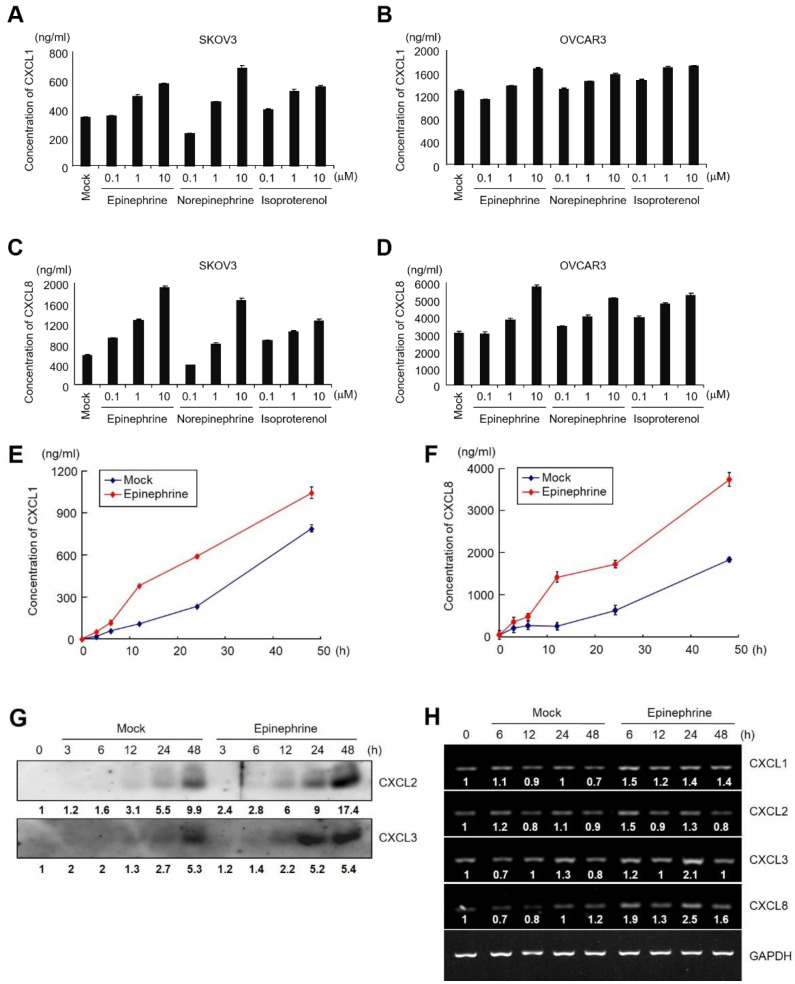
Catecholamines induce secretion of CXCL1, CXCL2, CXCL3, and CXCL8 in ovarian cancer cells. 1 × 10^6^ SKOV3 (**A**,**C**) or OVCAR3 (**B**,**D**) cells in 6-well plate were treated with 0.1, 1, or 10 μM epinephrine, norepinephrine, or isoproterenol. After 24 h, the concentrations of CXCL1 (**A**,**B**) and CXCL8 (**C**,**D**) in the culture medium were measured using ELISA. (**E**,**F**) Time-dependent secretion of CXCL1 or CXCL8 in SKOV3 cells. After 0, 3, 6, 12, 24, 48 h in the presence or absence of 10 μM epinephrine in SKOV3 cells, conditioned medium was obtained and the concentrations of CXCL1 (**E**) and CXCL8 (**F**) were measured by ELISA. (**G**) Levels of CXCL2 and CXCL3 were determined by Western blot analysis using anti-CXCL2 or CXCL3 antibodies. The time-dependent expression changes of the target gene are represented as fold change relative to the initial time (0 h). (**H**) The time-dependent expression changes of the target gene are represented as fold change relative to the initial time (0 h) and were normalized to GAPDH. (**H**) Epinephrine induces the mRNA expression of CXCL1, CXCL2, CXCL3, and CXCL8 in SKOV3 cells. 1 × 10^6^ SKOV3 cells in 6-well plate were treated with 10 μM epinephrine and cells were harvested after 0, 6, 12, 24, and 48 h. Total RNA was isolated and RT-PCR analysis was carried out for CXCL1, CXCL2, CXCL3, and CXCL8. GAPDH is shown as a control for RNA loading. Results are representative of two independent experiments.

**Figure 5 ijms-24-14104-f005:**
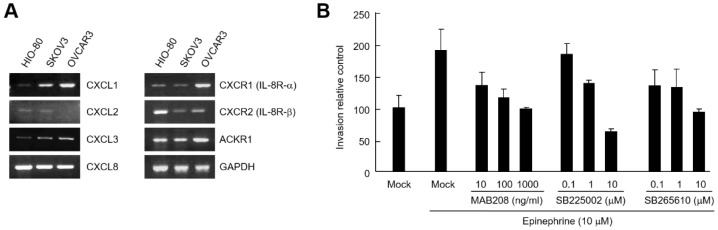
CXCL8 secreted by SKOV3 cells induces SKOV3 cell invasion. (**A**) mRNA levels of chemokines and chemokine receptors in ovarian cancer cells. Total RNA isolated from HIO-80, SKOV3, and OVCAR3 cells was analyzed by RT-PCR using chemokine (CXCL1, CXCL2, CXCL3, and CXCL8)- and chemokine receptor (CXCR1, CXCR2, and ACKR1)-specific primers. (**B**) 5 × 10^4^ SKOV3 cells were seeded into a matrigel-coated upper chamber with a lower chamber containing 10 μM epinephrine in the presence or absence of MAB208 (CXCL8 neutralizing antibody), SB225002, or SB265610. After culturing for 24 h, infiltrated cells were fixed, stained, and observed using an optical microscopy. Results are representative of two independent experiments.

**Figure 6 ijms-24-14104-f006:**
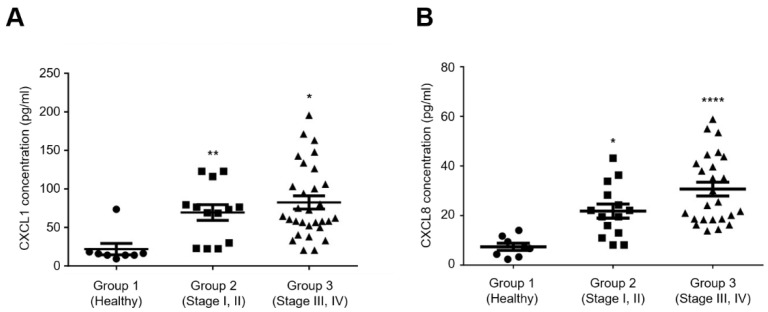
Elevated levels of CXCL1 and CXCL8 in plasma samples from ovarian cancer patients. Scatter plots show the concentrations of CXCL1 (**A**) and CXCL8 (**B**) in plasma samples from ovarian cancer patients at each stage of cancer. Analysis of CXCL1 and CXCL8 in plasma samples from ovarian cancer patients and normal blood donors were measured using ELISA. The horizontal bars represent the median and whiskers indicate the interquartile range. The healthy patients were used as the control group for statistical analysis. The *p*-value was calculated using one-way ANOVA, **** *p* < 0.0001, ** *p* < 0.01 and * *p* < 0.05.

**Table 1 ijms-24-14104-t001:** Table of primer sequences.

Primer Name (or Gene Name)	Primer Sequences 5′-3′	
	Sense	Antisense
Human α1A adrenergic receptor	GAAGGGCAACACAAGGACAT	CAGGAGGATTGGTCTTTGGA
Human α1B adrenergic receptor	CCTGAGGATCCATTCCAAGA	CGGTAGAGCGATGAAGAAGG
Human α1D adrenergic receptor	AGAAGAAAGCGGCCAAGACT	GCTGGAACAGGGGTAGATGA
Human α2A adrenergic receptor	CTACGTGCGCATCTACCAGA	AGACGAGCTCTCCTCCAGGT
Human α2B adrenergic receptor	TTCTTTGCTCCTTGCCTCAT	ACTTCGAGTGTCCGTTGACC
Human α2C adrenergic receptor	TCCGTCGAGTTCTTCCTGTC	GCTGAAGAAGAAGGGGAACC
Human β1 adrenergic receptor	CTGCTACAACGACCCCAAGT	GCAGCTGTCGATCTTCTTCA
Human β2 adrenergic receptor	CTGCGCAGGTCTTCTTTGA	CTTGATGGCCCACAAAGTCT
Human β3 adrenergic receptor	TCTTCTCGTGATGCTCTTCG	ATGAGACCCAAGGTGCACAG
Human CXCL1 (GRO-α)	AGTCATAGCCACACTCAAGAATG	TGGCCATTTGCTTGGATCCGC
Human CXCL2 (GRO-β)	CGCCCAAACCGAAGTCATAGC	GATGCTCAAACACATTAGGCGC
Human CXCL3 (GRO-γ)	CGCCCAAACCGAAGTCATAGC	ATCAAGGTGGCTGACACATTATGG
Human CXCL8 (IL-8)	GTGCAGTTTTGCCAAGGAGT	CTCTGCACCCAGTTTTCCTT
Human CXCR1	CCACCTGCAGATGAAGATTACA	AGGGACAGATTCATAGACAGT
Human CXCR2	CCAGAATCCCTGGAAATCAA	ACAGGCAGGGCCAGGAGCAA
Human ACKR1	GTCTTGTTGCCATTGGGTTT	GACAACAGCAACAGCTTGGA
Human GAPDH	CGGGAAGCTTGTGATCAATGG	GGCAGTGATGGCATGGACTG

## Data Availability

Datasets and materials are available upon reasonable request.

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
