# Peer review of "Catecholamines Promote Ovarian Cancer Progression through Secretion of CXC-Chemokines"

_ijms, 2023, doi:10.3390/ijms241814104_

Round 1

Reviewer 1 Report

The overall quality of the manuscript is good and has significant implications. However, it can be improved by incorporating the following major revisions.

·      In the introduction section, the existing gap that this study is aimed to fill is not clearly indicated. I suggest the authors identify a potential gap regarding the topic and include it in the introduction and explicitly state how this study will address it.  

·      The results and inferences from the study need not be provided in the introduction. 

·      Authors could consider rewriting the last paragraph of the introduction with the research question or hypothesis and briefly stress the rationale behind it. Although the hypothesis has been included in the discussion, introduction would be a appropriate to include it initially

·      Why have the authors included the method section after the discussion? Kindly follow the IMRAD format for writing the manuscript

·      The methodology should accurately describe the study design. If your approach is molecular docking, kindly state this clearly in the methods

·      Lines [302-335]: Mention what is described in that for readers to understand. Consider representing it in a table or chart with appropriate footnotes.

·      Mention the reason behind the steps stated in the lines [342-344]

·      The authors state that the number of cells were manually measured [line 345]. In such a case, consider including a statement for the possibility of any visual or counting errors. 

·       How was the overall analysis done and how is the data represented? The methods section should also include the methods of data analysis and representation.

·      [Line 402] In the conclusion, the authors have  mentioned that the study aimed to identify, “factors that increase the progression of ovarian cancer”. This statement completely changes the objective of the research. This study evaluated

The role of catecholamines and not generally any factors Using the term “factors” may include any physical, metabolic, molecular, habitual and even emotional factors that can potentially aggravate ovarian cancer

·      In the conclusion, mention how the results have addressed your research question or hypothesis.

·      Authors should include the limitations of the study, strengths, and directions of future research. These components will further add to your CV in the future. 

Minor english corrects are needed

Reviewer 2 Report

The study by Kim et al explores the role of adrenoceptor mediated chemokine secretion in ovarian cancer. The authors argue that stress signalling promotes chemokine secretion and this promotes ovarian cancer progression. This is one of the most well written manuscripts I have reviewed in a long time. Unfortunately, though, there are major issue with the underlying science that need to be addressed and this will require significant further experimentation.

Major

1.       Unfortunately, it has been known for some time that SkOV3 cells are probably not high grade serous ovarian cancer. https://www.nature.com/articles/ncomms3126. However they may be clear cell ovarian cancer https://genomemedicine.biomedcentral.com/articles/10.1186/s13073-021-00952-5. The authors should be clear about this in their manuscript.  The authors address this issue somewhat in figure 4 by using ovcar-3, which are a better model of HGSOC, but the changes (eg fig 4)  are much less pronounced than with the SKOV3 cells. The authors need to discuss the relevance of their findings.

2.       The concentrations of the adrenoceptor agonists are rather high compared to the EC50 reported in other studies. Are the authors sure they are acting through their cognate receptors? Studies with studies with selective antagonists are needed to confirm this. Otherwise, its plausible, if not highly likely, that the agonists are acting through another receptor.

3.       Fig 1B, fig 4 needs statistical analysis to show the changes are significant. All the figures need to state how many times the experiments were done. This need to make clear whether they are biological replicates (eg with a different passage of the cell line) or technical replicates within one experiment. The error bars (what are these, S.D, please state in the figure legend) in several figures are so small that I doubt these are biological replicates. If they are technical replicates from one experiment, the experiment needs repeating a further two times at least and the average data from three experiments presented. Fig H and all the other RT PCR experiments in the manuscript, need quantifying, including normalization to GAPDH and  (and ideally so does fig G)  the results from different biological replicates should be presented.

4.       Figure 2. I am a bit uncomfortable with this figure because I am shocked that the level of secreted proteins is so much higher than the background which I would have anticipated from the FCS in the medium. I appreciate that the authors used serum free medium, but I would have expected at least some residual protein. Can the authors comment.  What was the vehicle used for adding the drugs – was it also serum free medium?

5.       Figure 5 – why did the authors not look at CXCR1 antagonists/siRNA  and an antibody against CXCL1? This analysis seems incomplete.  Also the data seems to contradict fig 4H. In one figure the basal levels of CXCL8 are lowe than those of CXCL3 (fig 4) whereas in Fig 5 it’s the other way around.  Were identical numbers of cycles of amplification/amount of template the same ?

Minor

1.       Is the data in fig 3B same as in 3A but presented side by side to facilitate interpretation? Please clarify this in the figure legend

2.       Fig 6 – for clarlty – presumably the statistical comparison is to plasma from healthy patients – please state this in the figure legend.

Round 2

Reviewer 1 Report

I thank the authors for incorporating all the revisions mentioned. NO further revisions would be needed to qualify the manuscript for publication.